# In Situ Simultaneous Analysis of Nitrogen and Phosphorus Migration in Urban Black Odorous Runoff

**DOI:** 10.3390/ijerph192013240

**Published:** 2022-10-14

**Authors:** Ying Chen, Yu Yao, Xiaoxiang Han, Dujun Li, Ruiming Han

**Affiliations:** 1School of Environment, Nanjing Normal University, Nanjing 210023, China; 2Jiangsu Center for Collaborative Innovation in Geographical Information Resource Development and Application, Nanjing 210023, China; 3Jiangsu Engineering Laboratory of Water and Soil Eco-Remediation, Nanjing Normal University, Nanjing 210023, China

**Keywords:** black odorous, urban runoff, diffusive gradients in thin films (DGT), nitrogen, phosphorus

## Abstract

The extremely serious urban runoff eutrophication and black odorous phenomenon pose a significant threat to the lake aquatic ecosystem, resulting in a significantly increased frequency, magnitude, and duration of algal blooms in lakes. However, few investigations focus on small tributaries of the lakes, despite the ubiquity and potential local importance of these runoffs. Thus, the labile sediments NH_4_^+^-N, NO_3_^−^-N, PO_4_^3−^, Fe^2+^, and S^2−^ in black odorous runoff at Wuxi were overall analyzed at high resolution using diffusive gradients in thin films (DGT). The variations in labile N, P, Fe, and S distribution profiles at different sampling sites indicated high heterogeneity in sediments. The concentrations of labile P, Fe, and S showed synchronous variation from the sediment-water interface (SWI) up to −20 mm along sediment profiles. Moreover, there existed a significant positive correlation among labile P, Fe, and S concentrations (*p <* 0.05), which might represent typical odor compounds’ FeS and H_2_S synchronous release process in urban runoff. Furthermore, the apparent diffusion fluxes of labile P, Fe, and S across the SWI were all released upward, while fluxes of NH_4_^+^-N and NO_3_^−^-N release downward, indicating the sediments act as source and sink of P and N, respectively. Sediments’ potential for endogenous P and N fractions release results in the black−odorous water, and sediment finally abouchement the Taihu, which intensifies further lake eutrophication phenomenon.

## 1. Introduction

Urban runoffs play an essential role in the natural biochemical cycle, water utilization, and disposal [1]. However, with rapid urbanization and industrial development, numerous black odorous urban runoffs are increasing uncontrollably, posing serious risks to both the physical and mental health of surrounding people. Black odorous runoffs have been reported in some developed or developing countries, and data released in December 2019 showed that the number of black odorous runoffs in China had reached 2869. Even malodorous urban runoff might trigger eutrophication in lakes [2]. However, few investigations have reported the main factors leading to the release of reduplicative black odorous. Sediments play a significant role in contaminants cycling in urban runoff, which might act as a sink or a source of target elements on account of migration direction across the SWI [3]. Thus, significant heterogeneity in the temporal-spatial resolution of sediments and diverse biochemical interactions among various fractions in sediments result in complications in quantifying the labile fraction concentrations [4,5].

Nitrogen (N), phosphorus (P), iron (Fe), and sulfur (S) are essential nutrients for living organisms. Excessive accumulation of exogenous N and P fractions in runoffs could cause harmful algae blooms and perturb relevant elemental cycles [4,6,7], resulting in serious impacts on aquatic ecosystems [8]. Sediments produce or maintain a black odor mainly due to their capacity to consume oxygen and release various oxygen-consuming compounds into the water column. The typical reduction fractions (Fe and Mn) are widely accepted as the triggers that either inhibit or stimulate the labile N and P fractions’ mobility [9,10]. Under aerobic conditions, labile arsenic fractions tend to deposit in sediment because of the influence of Fe/Mn–(OOH) adsorption or coprecipitation processes. Fe/Mn–(OOH) transforms to Fe(II) and Mn (II) immediately once dissolved oxygen (DO) decreases below zero, resulting in N and P release to the overlying water because of the sorption sites’ loss of As [11]. Fe(II) can react with S (II) to form ferrous sulfide (FeS), which is reported to be the main contributor to the blackening of waterbodies. In addition, biogeochemical zones where migration (accumulation and release) and transformation (oxidation–reduction reactions) processes occur are liable to experience small variations (either at the millimeter or micrometer scale), most of which occur in the vicinity of the SWI [12]. Therefore, studying N, P, Fe, and S distributions in sediments and estimating their diffusion fluxes at the SWI are significant in understanding the mechanism of blackening and odor formation in sediments.

However, traditional ex situ techniques have defined bioavailable N and P fractions, tending to ignore the dynamic character that leads to the repeated dissolution and adsorption of N and P fractions and thus not reflecting the real concentration of labile P and N fractions under extreme high P and algal biomass stress [13]. Accordingly, in situ characterization of P and N bioavailability using robust techniques that obtain labile values under conditions of complicated physicochemical properties is necessary to understand the overall process of mobility variation of target fractions [14]. Thus, the typical in situ technique using diffusive gradients in thin films (DGT) technique is a potentially robust tool that is applied to simultaneously track the information for N, P, Fe, and S and sediment profiles. DGT could dynamically obtain the preconcentrate high-resolution information of target organic or inorganic diffusing species [15,16,17,18]. On the basis of Fick’s first law, the DGT technique remedies the limitations of ex situ, i.e., contamination speciation and concentration variation during the sampling process, by in situ characterization. The diffusive and specific binding gels are nested into DGT devices as the binding phase could selectively accumulate target fractions and thus constrain the accumulation speed [19]. Binding gels were initially cast using one or two different agents, aiming for one or two types of element accumulation [20]. To optimize the technique, a newly mixed binding gel, which contained ZrO and Chelex, was produced for simultaneous accumulation, i.e., oxyanions as well as metallic cations [4]. AMP-TH DGT was developed for simultaneous measurements of NH_4_^+^-N and NO_3_^−^-N [17,21].

In this study, we performed in situ simultaneous measurement of N, P, Fe, and S in sediment from typical black odorous runoff in Wuxi via the DGT technique, explored the relationships of N and P with S and Fe, and analyzed the apparent diffusion fluxes of contaminants across the SWI. Furthermore, microbial community composition and diversity were quantified by 16S rRNA analysis, aiming to (1) reveal the vertical distribution and the correlation of labile P, labile Fe, labile S, labile NH_4_^+^-N, and labile NO_3_^−^-N in urban black odorous runoff; (2) reveal the main factor for urban black odorous runoff and eutrophication; (3) clarify the effect of labile Fe and S on the migration of N and P.

## 2. Materials and Methods

### 2.1. Study Area

On account of its location, Wuxi has a dense water network and intertwined runoff. Numerous runoffs in Wuxi were significantly affected by industrial, agricultural, and domestic wastewater for a long period, accumulating a large number of external contaminants, resulting in the black odorous runoff, as well as the seriously deteriorative environmental ecosystem of Taihu. However, there exist significantly few reports on contaminants distributions and transformation in small-sized and midsized tributaries of the Taihu. Thus, PuShe River (PSH) and NeiTang River (NTH) in Liangxi District, and DaCheng Branch (DCB) in Hubin District of Wuxi, were selected as representative of the urban black odorous runoff (Appendix A). The locations of the sampling sites are shown in Figure 1. The details of the longitudes and latitudes of the selected sites are shown in Appendix A.

### 2.2. Field Sampling

Flat-type AMP-TH DGT and flat-type ZrO-CA DGT probes were used to simultaneously measure NH_4_^+^-N and NO_3_^−^-N, as well as labile Fe, P, and S concentrations, respectively. (EasySensor Ltd., Nanjing, China). Each DGT device was equipped with binding gel, agarose diffusive gel, and PVDF membrane. However, different types of DGTs were assembled with different binding gels, AMP-TH DGT equipped with AMP-TH resin gel and ZrO-CA DGT device equipped with ZrO and Chelex-100 resin gel, respectively. The sampling window of each device is 15 × 2 cm^2^. All probes were deoxygenated with N_2_ for at least 16 h and were sealed in containers that were filled with deoxygenated 0.01 M NaCl solution prior to in situ deployments.

Two kinds of probes were placed, attached to buoys, and vertically inserted into the sediment by the deployment device to ensure the probes were approximately 2–4 cm above the SWI. The DGT probes were deployed for 24 h. Then, the particles adhered to the DGT surface were quickly rinsed with deionized water after retrieval to prevent the re-proliferation of the target elements. The DGT devices were then placed in a self-sealing bag with a few drops of deionized water to ensure the moisture environment before the laboratory analysis [22]. Following the in situ measurements, the surface runoff sediments were collected using a Peterson grab sampler and frozen at −80 °C in an ultralow-temperature freezer for 16S rRNA amplicon sequencing.

### 2.3. Sample Analysis

The AMP-TH resin gels and ZrO-CA resin gels were disassembled from the deployed DGT probes. Each AMP-TH resin gel was vertically and evenly sliced at a 2 mm interval using a cutter made by stacking ceramic blades with a thickness of 1 mm (Easy Sensor Ltd., Nanjing, China). Each resin slice was eluted with 0.8 mL of 1 M NaCl for 24 h. The concentration of NH_4_^+^-N and NO_3_^−^-N were determined by Nessler’s reagent colorimetric method and ultraviolet spectrophotometric method, respectively [23]. Microvolume eluents were measured using a 96-microplate spectrophotometer with Spark 10 M microplate reader (Tecan, Austria) [24].

Two-dimensional images of labile S were obtained with the computer-imaging densitometry (CID) technique at the submillimeter scale. The resin gels were scanned using a flatbed scanner (Cannon 5600F, Beijing, China) at a resolution of 600 dpi. ImageJ 1.46 was employed to analyze the grayscale intensity of the scanned images [6]. The resin gels were then vertically and evenly cut into 2 mm slices by the multibladed ceramic cutter previously described, which were transferred into 0.8 mL 1 M HNO_3_ for 16 h for labile Fe fractions extraction. Then, resin gel was added in a centrifuge tube with 0.8 mL 1 M NaOH for 24 h for labile P fractions extraction. Labile Fe and P fractions were determined by the o-phenanthroline colorimetric method and the phosphomolybdenum blue colorimetric method, respectively [25,26]. Microvolume eluents were measured using the 96-microplate spectrophotometer method with Spark 10M microplate reader [24].

### 2.4. Data Proceeding

Labile NH_4_^+^-N, NO_3_^−^-N, Fe, and P concentrations are calculated using Equation (1):(1)$$C_{DGT}=\frac{M∆g}{DAt}$$

The *t* (s) is the deployment time of *DGT*. A is the surface area of each gel slice (cm^2^). *D* is the diffusion coefficient of the target elements in the diffusive layer (cm^2^ s^−1^). ∆*g* is the thickness of the diffusive layer (cm). *M* is the accumulated mass of target elements in the binding gel (mg), which is calculated using Equation (2) [27,28]:(2)$$M=\frac{C_{e}V_{e}}{f_{e}}$$
where $c_{e}$ is the concentration of elution solution (mg L^−1^), $V_{e}$ and $f_{e}$ are the volumes of elution solution (mL) and elution efficiency, respectively.

The apparent flux (F, mg m^−2^ d^−1^) at the SWI was calculated following Equation (3):(3)$$F=F_{W}+F_{S}={\left(-D_{W}\frac{\partial C_{w}}{\partial X_{W}}\right)}_{x=0}+{\left(-\varphi D_{S}\frac{\partial C_{s}}{\partial X_{S}}\right)}_{x=0}$$
where F_W_ and F_S_ represent the net diffusive fluxes of target elements DGT labile concentration of the overlying water and the sediment (mg m^−2^ d^−1^), respectively, $\frac{\partial C_{w}}{\partial X_{W}}$ and $\frac{\partial C_{s}}{\partial X_{S}}$ are the concentration gradients of the overlying water and the sediment, *C* is the concentration of target elements (mg L^−1^), and x is the depth (cm). *D_W_* and *D_S_* represent the effective diffusion coefficients in water and sediment, respectively, and is the porosity, which is calculated using Equation (4) [28,29]:(4)$$\varphi =\frac{\left(W_{w}-W_{d}\right)\times 100\%}{\left(W_{w}-W_{d}\right)+W_{d}/\rho }$$
where $W_{w}$ is the wet weight of sediments (g); $W_{d}$ is the dry weight of sediments (g); ρ is the ratio of the average density of surface sediment to water density (generally taken as 2.5).

### 2.5. S rRNA Amplicon Sequencing

Amplification sequencing of 16S rRNA was performed by Genesky Biotechnologies Inc., Shanghai (China). Briefly, total genomic DNA was extracted using the FastDNA^®^ SPIN Kit for Soil (MP Biomedicals, Santa Ana, CA, USA) according to the manufacturer’s instructions. The integrity of genomic DNA was detected through agarose gel electrophoresis, and the concentration and purity of genomic DNA were detected through the Nanodrop 2000 and Qubit3.0 Spectrophotometer. The V3-V4 hypervariable regions of the 16S rRNA gene were amplified with the primers 341F (5′-CCTACGGGNGGCWGCAG-3′) and 805R (5′-GACTACHVGGGTATCTAATCC-3′) and then sequenced using Illumina NovaSeq 6000 sequencer.

### 2.6. Statistical Analysis

The 2D and 1D spatial distributions of the DGT fluxes and concentrations across the SWI were plotted with OriginPro 2018C 64Bit (OriginLab Inc., Northampton, MA, USA). SPSS 22 for Windows (SPSS Inc., Chicago, IL, USA) was used for statistical analysis. Pearson correlation analyses were performed to explore any underlying relationships. For all analyses, results were considered statistically significant when *p* < 0.05. Based on OTUs data, alpha diversity and Microbial community composition were calculated with QIIME 2 and displayed with R software. The metabolic prediction was analyzed using the functional annotation of the FAPROTAX database based on the 16 S rRNA gene data to explore the biogeochemical cycle functions of microorganisms (http://cloud.geneskybiotech.com/index.html (accessed on 17 June 2021). The community dissimilarities between different sampling sites based on Bray–Curtis distances were also measured. Abiotic factors that affected the microbial community variation were evaluated by the BIOENV. The predominant environmental factors controlling the variation of the microbial community in sampling sites of runoff were analyzed using redundancy analysis (RDA) in the R package vegan.

## 3. Results

### 3.1. Physicochemical Properties of Overlying Water

The physicochemical properties of the overlying water are shown in Table 1. The pH value was in a range of 7.29–7.65, showing a weakly alkaline. The OPR at all sampling sites ranged from 251.4 to 284.9 mV. The concentration of DO value varied from 3.92 to 6.02 mg L^−1^, with the highest at DCB-2 and the lowest at NTH, reflecting an oxic condition in the benthic bottom. The concentrations of labile Fe, P, and S in the overlying water varied considerably at different sites and were measured ranging from 0.072 to 0.39 mg L^−1^, from 0.25 to 6.71 mg L^−1^, and from 0.0025 to 0.011 mg L^−1^, respectively. Labile NO_3_^−^-N and NH_4_^+^-N ranged from 7.26 to 24.29 mg L^−1^ and 4.00 to 44.78 mg L^−1^, respectively. High concentrations of labile NO_3_^−^-N and NH_4_^+^-N indicated the extremely serious situation of urban runoffs, accounting for industrial and domestic wastewater drainage.

### 3.2. Variations of Labile N and P in Vertical Profiles

The distribution of labile NH_4_^+^-N and labile NO_3_^−^-N profiles are shown in Figure 2a. The mean concentration of labile NH_4_^+^-N in sediment profiles ranged from 4.67 to 38.92 mg L^−1^. Labile NH_4_^+^-N concentration showed no obvious variations across the SWI at PSH-1, DCB-2, and DCB-3. However, NH_4_^+^-N significantly increased along the profile in the range of 0 to −10 mm at PSH-2 and decreased along the profile in the range of 0 to −10 mm and 10 to 0 mm at DCB-1 and NTH, respectively. It is worth noting that an extremely high concentration peak of labile NH_4_^+^-N (47.07 mg L^−1^) appeared at a depth of −52 mm at NTH; on account of this, high heterogeneity may be caused by bioturbations. For the labile NO_3_^−^-N, the concentration at all sampling sites displayed a clear diffusion gradient across the SWI. The mean labile NO_3_^−^-N concentration in sediments ranged from 6.89 to 7.72 mg L^−1^, which was lower than that in the overlying water. It was also clear that the extremely low concentration of labile NO_3_^−^-N (0.29 mg L^−1^) exited at a depth of −22 mm at PSH-2. Furthermore, the concentration of labile NO_3_^−^-N in the sediment profiles of NTH increased (from 3.92 to 11.96 mg L^−1^) along the profile; however, there were no significant variations in other sediment profiles.

The variations of labile P in sediment profiles are shown in Figure 2b. The labile P at all profiles varied from 0.15 to 16.19 mg L^−1^ and was low and did not vary significantly with depth above the SWI, while dramatically increased with depth in the range of 0~−20 mm of profiles. Labile P concentration at PSH-1, DCB-1, DCB-2, DCB-3, and NTH sediment profiles maintained slight fluctuation or increased after significantly increasing to maximum values of 9.18, 15.42, 13.99, 9.13, and 8.27 mg L^−1^, respectively. Labile P concentration decreased gradually with depth after reaching a maximum of 16.19 mg L^−1^ in the vicinity of SWI at PSH-2. In addition, the labile P concentrations in sediment profiles at DCB-1, DCB-2, and DCB-3 were slightly higher and ranged from 6.20 to 16.15 mg L^−1^, while the other three sites ranged from 2.04 to 9.79 mg L^−1^. Furthermore, as Figure 2b shows, the concentration of labile Fe at all sites varied from 0.015 to 1.54 mg L^−1^. The labile Fe concentration significantly increased at the SWI. The obvious synchronization of labile P and labile Fe was observed in the range of 0~−20 mm of profiles. Moreover, the concentration of labile S increased sharply to the peak in the range of 0~−20 mm of profiles; the peaks were recorded in both one-dimensional and two-dimensional distributions of labile S (Figure 3). These results indicated the simultaneous release of P, Fe, and S.

### 3.3. Microbial Community Composition of the Runoff Sediment

The diversity index of the microbial community is shown in Table 2. The Chao 1 index of all sampling sites is in the descending order: NTH (2501.45) > PSH-2 (2444.56) > PSH-1 (2391.95) > DCB-3 (2116.41) > DCB-2 (1869.13) > DCB-1 (1656.93). The Shannon index of all sampling sites followed the order across sites: NTH (7.71) > PSH-2 (6.99) > PSH-1 (6.94) > DCB-3 (6.70) > DCB-1 (6.49) > DCB-2 (6.31). The Good’s coverage was greater than or equal to 99.9%, suggesting that this research covered the majority of the microbial community with sufficient sequence coverage, and the results could represent the real microbial situation in the sediment. In addition, the microbial community composition of runoff sediments at the phylum level was determined (Figure 4). The results showed that the composition varied considerably among six sediment samples. Proteobacteria was the most abundant phylum of the communities, representing 40.25~57.79% of the general microbial community, with the highest relative abundances at DCB-2 and lowest at DCB-1. The *Chloroflexi*, *Firmicutes*, and *Bacteroidetes* were also the dominant communities, accounting for 9.52~19.41%, 4.43~12.80%, and 2.62~10.00% of total sequences, respectively. Meanwhile, some no-rank species were detected, which could result in a minor underestimation of the actual distribution. Furthermore, the PCoA (Appendix A) clearly revealed that DCB-1 had a unique species composition compared with the other sampling sites, whereas DCB-2 and DCB-3 had the most comparable species composition. Eventually, the sediment microbial functions were predicted using FAPROTAX based on amplicon sequencing data. The top 30 most abundant identified microbial biochemical functions are shown in the heatmap (Appendix A), including chemoheterotrophy and aerobic chemoheterotrophy as the most common functions, and the predicted microbial functions involved in chemoheterotrophy, fermentation, anaerobic respiration (such as nitrogen, nitrate, nitrite, sulfur, sulfur compound, iron), nitrification, and hydrocarbon degradation. Overall, the predicted microbial functions were closely related to the nitrogen, sulfur, and iron cycles in urban black odorous runoff sediments.

## 4. Discussion

### 4.1. Apparent Diffusion Flux of Labile N, P, Fe, and S across the SWI

The apparent diffusion flux of labile N, P, Fe, and S across the SWI of six sites is presented in Figure 5. The apparent diffusion flux characterized the extent and direction of contaminants across the SWI. The positive (negative) flux indicated the upward (downward) release of contaminants [17,30,31]. Except for labile Fe at DCB-3, significant positive values of the diffusion fluxes of labile P, Fe, and S were determined. The results were comparable to the apparent diffusive flux of P, Fe, and S in the Peal River Delta region’s black odorous runoff [32]. As shown in Figure 5, the positive apparent diffusive flux of P was high (ranged from 10.31~27.26 mg m^−2^ d^−1^), which indicated that the runoff sediments play the role of a source of P and had a high capacity to release P into the overlying water via SWI. The release of P aggravated the P load in the overlying water. Moreover, the high internal P load was considered to be the major factor that delayed ecosystem recovery after the reduction in external P inputs [33]. In contrast to the labile P diffusion flux trends, negative diffusion fluxes of labile NH_4_^+^-N (ranged from −0.91~−10.55 mg m^−2^ d^−1^) and NO_3_^−^-N (ranged from −1.04~−36.84 mg m^−2^ d^−1^) were found. The negative fluxes indicated that the runoff sediments play the role of a nitrogen sink, and the sediments had a large capacity to store NH_4_^+^-N and NO_3_^−^-N. These are all attributed to the fact that there existed active nitrification and denitrification microbial processes in the sediments; NH_4_^+^-N and NO_3_^−^-N in sediments were reduced to N_2_. Then, NH_4_^+^-N and NO_3_^−^-N in the overlying water diffused downward the sediment. This diffusion process reduced the concentration of NH_4_^+^-N and NO_3_^−^-N in the overlying water, finally resulting in the removal of the N fraction from the aquatic ecosystem [34]. In general, the flux of sampling sites was different, which was determined by the concentration gradients of target elements in the overlying water and sediments, while the contaminant concentrations in the profiles exited significant differences. On the one hand, the sediments were highly heterogeneous. On the other hand, the sampling sites are located in residential areas, and domestic sewage of different properties may be discharged into the runoffs, resulting in a difference in the contaminants in the sediment profile of the runoffs.

### 4.2. Migration of Nitrogen and Phosphorus in Sediment

Internal loadings of N and P in runoffs include the major internal cycling at SWI and the minor algae-derived inner loading [8]. Internal loadings of N and P play an important role in eutrophication, while internal loadings of Fe and S are widely recognized as crucial elements directly related to the color and odor of the runoff. [35]. Overall, N, P, S, and Fe are inextricably associated with the quality of runoff waterbody.

Denitrification, anaerobic ammonium oxidation (ANAMMOX), and dissimilatory nitrate reduction to ammonia (DNRA) are crucial N cycling pathways in aquatic ecosystems [36,37]. Both S and Fe affected the N cycling in sediments. Previous studies have confirmed that labile S has obvious inhibitory effects on denitrification and promotes the DNRA on account of NO_3_^−^-N/NO_2_^−^-N that accumulated after the inhibition of denitrification, and sulfide that exited in sediments could be used by DNRA bacteria as an effective electron acceptor and donor, respectively [38,39,40,41]. In addition, sulfide is widely reported to be an electron donor for S–dependent denitrification (NO_3_^−^/S^2−^→N_2_/SO_4_^2−^(S^0^)) [42,43,44]. Negative relationship (*p* < 0.01 or *p* < 0.05) between labile NO_3_^−^-N and labile S at PSH-1, DCB-1, DCB-2, and DCB-3 (Table 3), inferring with the denitrification process, was indeed affected by S. Until now, whether labile sulfide promotes or inhibits ANAMMOX has long been a subject of the debate. However, Pearson correlation analysis showed a significant relationship between labile NH_4_^+^-N and labile S at PSH-2, DCB-1, and DCB-3 (*r*^2^ = −0.432, −0.566, and 0.616, respectively, *p* < 0.01), and no correlation at other three sites. The extent to which sulfide inhibited ANAMMOX was influenced by pH level and sulfide concentration, but the exact threshold was unclear up to now [40]. Thus, it is suspected that differences in substrate concentrations and environmental conditions at the sampling sites have led to such a result. Furthermore, Fe(III) reduction promotes ANAMMOX (termed Feammox) is a new pathway of N transformation that has been discovered recently [45]. Feammox can produce N_2_ through the reduction of Fe(III) and oxidation of NH_4_^+^-N (NH_4_^+^/Fe^3+^→N_2_/Fe^2+^) [46,47]. Significant positive relationships (*p* < 0.01) between labile NH_4_^+^-N and labile Fe were observed at DCB-2 and DCB-3.

The reduction and dissolution of Fe(III) oxyhydroxides is a primary process responsible for P release; the process promotes the release of Fe(II) into the overlying water [4,6]. Significant positive correlations (*p* < 0.01) between labile P and Fe(II) were observed at six sites (Table 3), demonstrating that the release of phosphorus in runoff sediment is controlled by Fe redox. In addition, except for the site of NTH, significant positive correlations (*p* < 0.01 or *p* < 0.05) between labile Fe(II) and S(-II) and labile P and S(-II) were also observed at the other five sites. The results indicated the simultaneous release of Fe(II) and S(-II) and P and S(-II). Specifically, iron reduction and sulfate reduction exist simultaneously under anoxic environments [13], where the reduction and dissolution of Fe(III) oxyhydroxides promote the release of P, and the sulfate reduction produces S(-II), resulting in the simultaneous release of P and S. Afterwards, the S(-II) combined with Fe(II) and formed black ferrous sulfide precipitates (FeS), which could explain the occurrence of black water in runoff [48]. Furthermore, the sulfate reduction process produced volatile gases, such as H_2_S, which contribute to the odor of the runoffs [35]. Although we observed the correlation between these elements in this study, to fully explain the key mechanisms of transformation and release of the contaminants, further studies should be conducted in conjunction with molecular biotechnology.

### 4.3. Effect of Environmental Variables on Labile N and P

RDA based on phylum level information from sediment samples was used to further reveal the relationships between labile N and P, microbial communities, and environmental factors (Figure 6). Total organic carbon (TOC) and DO were predominant for special bacterial and contaminants variations. The first axis of RDA1 was positively correlated with TOC, *Actinobactria*, *Firmicutes*, *Chloroflexi*, and labile NH_4_^+^-N, which explained 55.31% of the total variance. The second axis of RDA2 was positively correlated with labile NO_3_^−^-N and *Bacteroidetes*, explaining 16.74% of the total variance. The correlation results between N fraction and bacteria differ from a previous urban lake study [49], which could attribute to N fraction enrichment driving the shift of bacteria abundance and diversity in urban runoff sediments, and ultimately leading to the alteration of N transformation pathways [50]. It is well accepted that NH_4_^+^-N in pore water is mainly generated by the degradation of organic matter during early diagenesis [51,52]. The concentration of TOC in the runoff sediments ranged from 68.97 to 89.55 g/kg (Appendix A), which is significantly higher than in the other reported runoff (ranged from 16.02 to 41.49 g/kg) and Taihu (ranged from 8.50 to 16.38 g/kg) [53,54]. Therefore, the high TOC concentration in sediment explains the high levels of labile NH_4_^+^-N in the pore water of sediment profiles. Furthermore, DO and pH had a significantly positive effect on labile P, indicating that the DO and pH were the predominant controlling factors of P cycling in runoff [55,56]. It is reported that significant differences in the bacterial communities have been observed between urban and suburban sediment [48,57,58,59], which was the case with this investigation as well. Thus, the input of contaminants to urban runoff could affect not only the physicochemical properties of the overlying water but also the structure and diversity of bacterial communities and may further alter the ecological importance of specific communities in the runoff environments.

## 5. Conclusions

This study investigated the concentration distributions and diffusion flux of labile N P, Fe, and S in black odorous runoff sediments by using the DGT technique. The results showed that sediment profiles had a high concentration of both labile N and P, and black-odorous runoff sediment was not only the source of P, Fe, and S but also the sink of N. The concentrations of labile N, P, Fe, and S varied greatly between sampling sites, and the migration and release of N and P were influenced by Fe and S. The blackness and odorous of the runoffs are mainly due to the FeS and H_2_S. Moreover, TOC and DO were predominant for special bacterial and contaminant variations in urban runoff sediments. This study provides basic data for the management and control of black odorous runoff and lake sediments. Nonetheless, these data alone are insufficient, and more in-depth research needs to be addressed in future investigations. First, the seasonal climate has effects on the migration of target elements, and measurements of contaminants in black odorous runoff in different seasons are essential. Second, different regions of black odorous runoff have different levels of pollution, which could affect the microbial composition. Thus, the investigation and comparison of the different cities of black odorous runoff are necessary. Third, excessive contaminants input from urban runoff contributes to the increase in the frequency, magnitude, and duration of algal blooms in lakes. Most previous studies have focused on large receiving water bodies such as Lake Taihu; thus, the lack of attention given to contaminants input from numerous urban runoffs. Consequently, enhanced research on contaminants from urban runoffs is needed to quantify the environmental impact of urban runoffs on the large receiving lakes. Taking all these issues into consideration, a comprehensive understanding of the N and P migrations in black odorous runoff is required.

## Figures and Tables

**Figure 1 ijerph-19-13240-f001:**
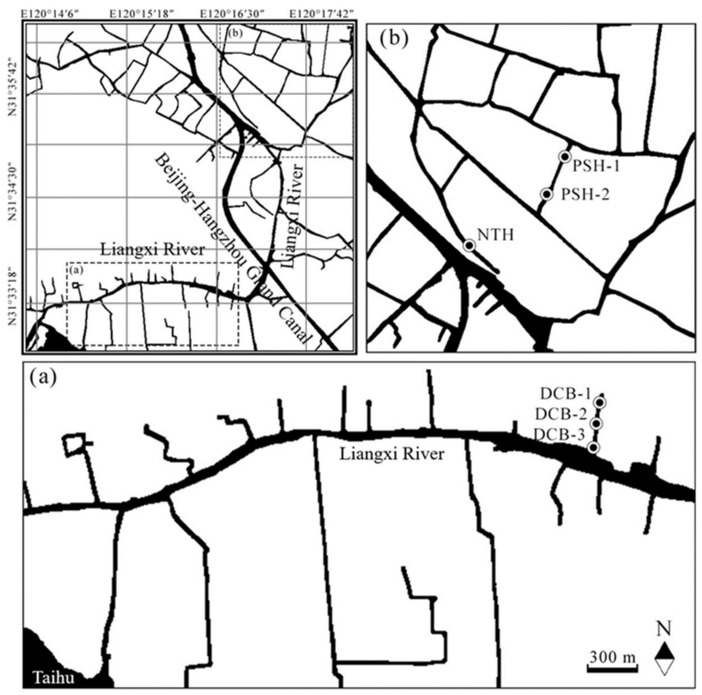
The distribution of sampling sites: (**a**) the distribution of DCB sampling sites; (**b**) the distribution of PSH and NTH sampling sites.

**Figure 2 ijerph-19-13240-f002:**
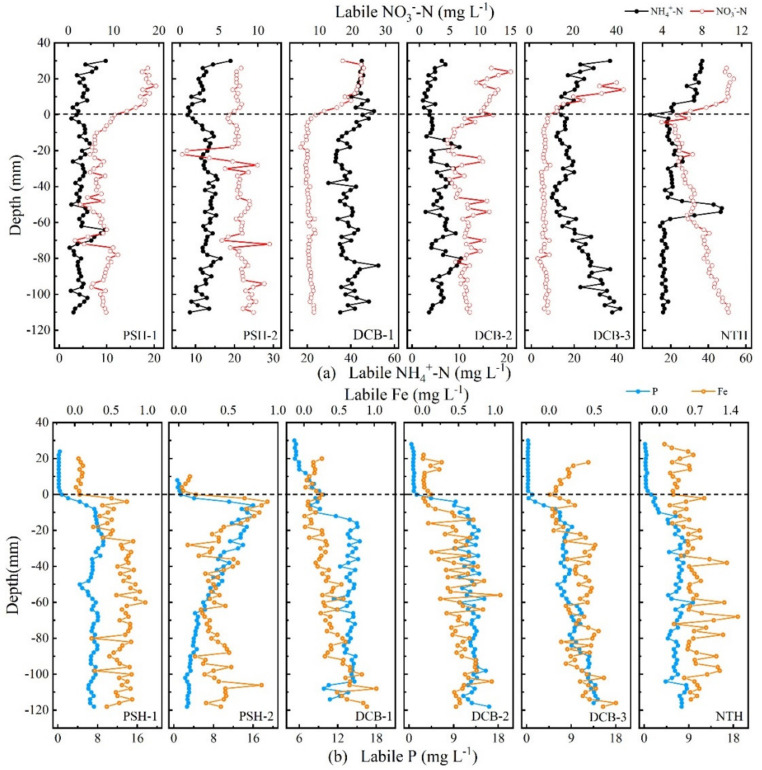
Distribution of labile NH_4_^+^-N and labile NO_3_^−^-N in sediment-water profile (**a**), and distribution of labile P and labile Fe in sediment-water profile (**b**). The dotted lines at a depth of zero show the position of the SWI.

**Figure 3 ijerph-19-13240-f003:**
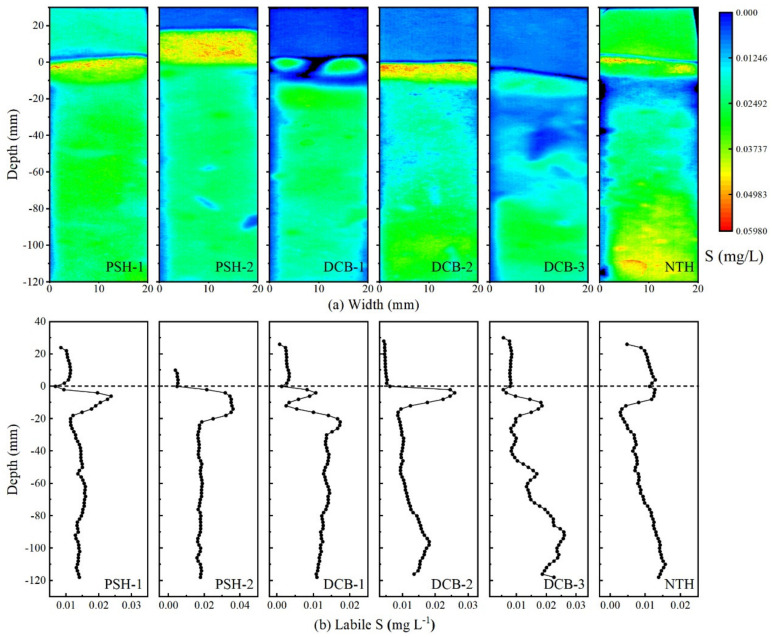
One-dimensional distribution (**b**) and two-dimensional distribution (**a**) of labile S. The dotted lines at a depth of zero show the position of the SWI.

**Figure 4 ijerph-19-13240-f004:**
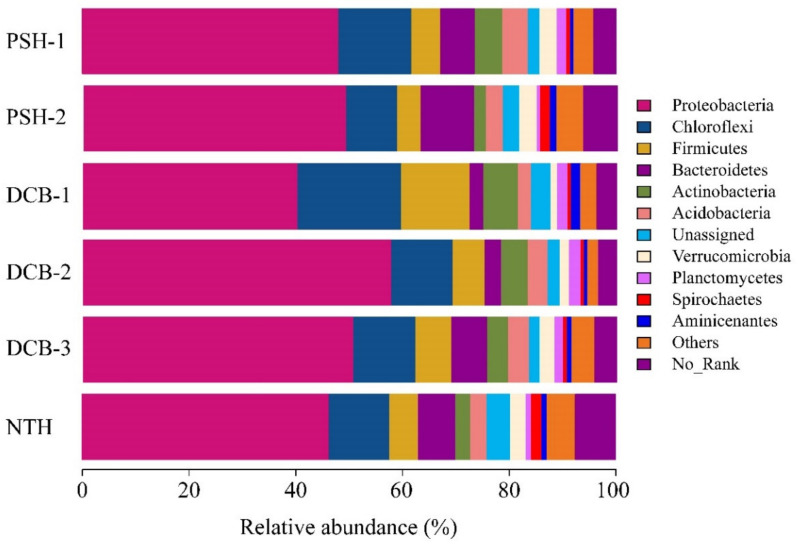
Relative abundance of sediment bacteria community composition at the phylum level.

**Figure 5 ijerph-19-13240-f005:**
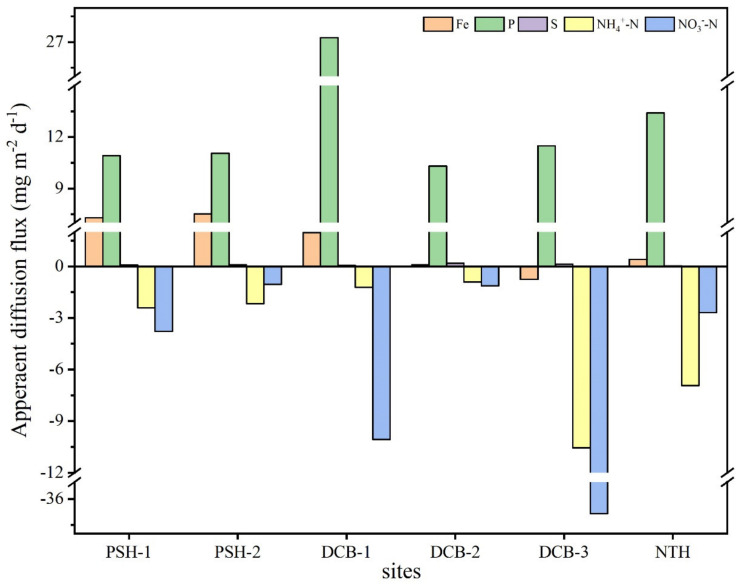
The apparent diffusion flux of the labile P, Fe, S, NH_4_^+^-N, and NO_3_^−^-N across the SWI.

**Figure 6 ijerph-19-13240-f006:**
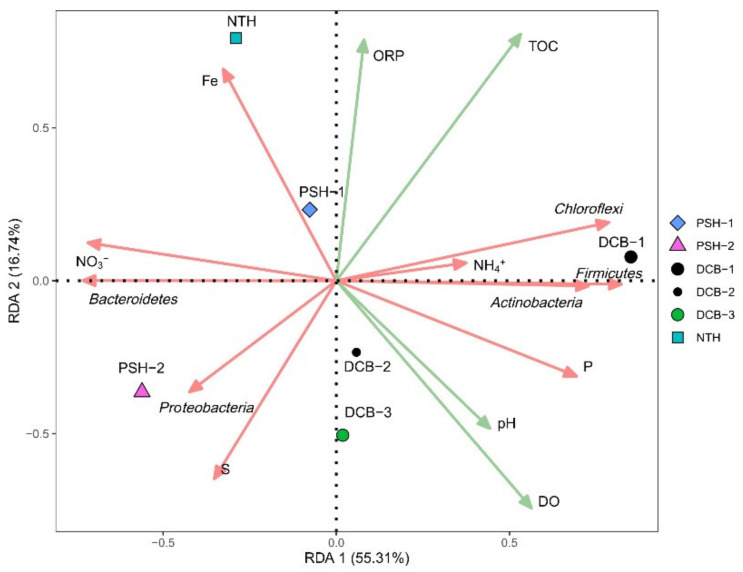
Redundancy analysis plot of the relationship among bacteria and dominant environmental parameters in sediment. Green solid lines indicate major environmental variables. Red solid lines show elements measured in sediments and the top five bacteria in relative abundance at the phylum level. Geometric graphs represent the sampling location.

**Table 1 ijerph-19-13240-t001:** Physicochemical properties of overlying water.

Sites	pH	ORP	DO	Labile Fe	Labile P	Labile S	Labile NO_3_^−^	Labile NH_4_^+^
		(mV)	(mg L^−1^)	(mg L^−1^)	(mg L^−1^)	(mg L^−1^)	(mg L^−1^)	(mg L^−1^)
PSH-1	7.29	281.1	4.20	0.072	0.25	0.010	16.24	5.27
PSH-2	7.53	270.0	4.87	0.087	1.02	0.0047	7.26	11.36
DCB-1	7.63	281.1	5.85	0.14	6.71	0.0025	19.17	44.78
DCB-2	7.65	251.4	6.02	0.11	0.89	0.0048	11.69	4.00
DCB-3	7.50	254.4	5.73	0.18	0.29	0.0078	24.29	20.15
NTH	7.41	284.9	3.92	0.39	0.41	0.011	9.75	29.86

**Table 2 ijerph-19-13240-t002:** The alpha diversity of the microbial sediment community.

Sites	Observed	Chao1	Shannon	Coverage
PSH-1	2379	2391.952	6.943	0.9989
PSH-2	2435	2391.952	6.985	0.9990
DCB-1	1650	1656.926	6.490	0.9994
DCB-2	1854	1869.128	6.318	0.9990
DCB-3	2103	2116.409	6.700	0.9990
NTH	2493	2501.448	7.105	0.9991

**Table 3 ijerph-19-13240-t003:** Pearson correlation values between labile Fe, P, S, NH_4_^+^, and NO_3_^−^.

Sample Site	Fe vs. P	Fe vs. S	Fe vs. NH_4_^+^	Fe vs. NO_3_^−^	P vs. S	P vs. NH_4_^+^	P vs. NO_3_^−^	S vs. NH_4_^+^	S vs. NO_3_^−^
PSH-1	0.703 **	0.281 *	−0.253 *	−0.739 **	0.517 **	−0.145	−0.866 **	−0.205	−0.471 **
PSH-2	0.497 **	0.732 **	−0.376 **	−0.138	0.703 **	−0.413 **	−0.404 **	−0.432 **	−0.059
DCB-1	0.531 **	0.480 **	−0.139	−0.411 **	0.922 **	−0.570 **	−0.858 **	−0.566 **	−0.826 **
DCB-2	0.788 **	0.452 **	0.334 **	−0.530 **	0.558 **	0.448 **	−0.681 **	0.153	−0.470 **
DCB-3	0.756 **	0.506 **	0.416 **	−0.118	0.793 **	0.423 **	−0.463 **	0.616 **	−0.269 *
NTH	0.556 **	0.134	−0.347 **	0.030	−0.095	−0.455 **	−0.171	−0.228	0.627 **

* and ** represent the correlation is significant at the level of *p* < 0.05 and *p* < 0.01, respectively.

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
