# Peer review of "In Situ Simultaneous Analysis of Nitrogen and Phosphorus Migration in Urban Black Odorous Runoff"

_ijerph, 2022, doi:10.3390/ijerph192013240_

Round 1

Reviewer 1 Report (Previous Reviewer 2)

The authors had listed the point-by-point responses to explain my original questions.

This manuscript is a resubmission of an earlier submission. The following is a list of the peer review reports and author responses from that submission.

Round 1

Reviewer 1 Report

The present study investigates the diffusion flux of chemicals/contaminants present in urban runoff across the sediment-water interface in selected small tributaries of a lake in Wuxi, China. The study also attempts to investigate the extent to which these sediments act as a source and/or sink of these chemicals, which can in turn contribute to increased pollution of the aquatic system. The manuscript likely makes a decent scientific attempt at the aforementioned. However, it currently suffers from significant issues; these need to be addressed urgently before publication can be recommended. These are detailed below.

-          The writing can be significantly improved, in particular paying attention to the English language – a few grammatical errors were noted throughout the manuscript and the writing was hard to follow in several instances; hence major revision of the English language of the entire manuscript is strongly recommended.

-          The background and motivation of the current study were not clearly stated in the introduction. The review of the literature relating to the subject area was inadequate and the novelty of the study was not very clear.

-          There is a major disconnect between the objectives of the study, discussion and conclusion.

-          More details on the methodology of the microbial community and sequencing analysis are required (section 2.6).

-          The contribution of the microbial community analysis to the discussion and interpretation of the processes driving contaminant release across the SWI was missing – it was thus not clear how the microbial community results add to the process discussion. Moreover, results of the microbial community were only presented at phylum level – this category presents a broad range of microbes exhibiting a diverse range of metabolic functioning – discussion at lower levels (e.g. at least order or family levels) is thus recommended.

-          The difference in contaminant concentrations within the sediment profile across the different locations/tributaries (e.g. Figure 2, Figure 3) was not discussed.

-          While the correlation results presented here (e.g. Table 3) are important in unravelling the likely mechanisms driving contaminants transformation and release, I think that they cannot be used to fully explain key mechanisms – but can also be used to infer potential mechanisms that would then inform future studies – I think that this should be more clearly stated in the discussion section.

-          The conclusion section should more clearly state the main conclusions of the study based on the results.

Reviewer 2 Report

 With current figure 1, S1 and table 1, it is difficult to figure out the relationship between sampling sites and Taihu/city. The figure 1 is recommended to improve.

Beside the overall sites that were limited, and it was also unknown how many replicates were made for each index in every site, thus the conclusion of the paper could be very weak. So the authors are recommended to add more explanation on this aspect, at least how many replications, if the number of sampling sites could not be increased. In sequence, data listed as mean ± S.D or mean ± S.E. would be better than single mean.

For the data of microbial community, the authors didn’t clearly explained what kind of samples were prepared for analysis. Also more explanation was needed to clarify how these date could support the main opinion of the paper. 

Round 2

Reviewer 2 Report

For question 1, the authors have added some marks in Fig. 1 in the revised version, which shows the relationship between sampling sites and taihu Lake.

For question 2, I haven’t seen any improvement on this aspect in the revised version, and still, the current data are too weak to support the conclusion.

For question 3, the authors had added explanation on sample collection for microbial analysis, whereas still little information was given on the importance of these data in supporting the main conclusion.